# $\alpha$VIL: Learning to Leverage Auxiliary Tasks for Multitask Learning

## Abstract

Multitask Learning is a Machine Learning paradigm that aims to train a range of (usually related) tasks with the help of a shared model. While the goal is often to improve the joint performance of all training tasks, another approach is to focus on the performance of a specific target task, while treating the remaining ones as auxiliary data from which to possibly leverage positive transfer towards the target during training. In such settings, it becomes important to estimate the positive or negative influence auxiliary tasks will have on the target. While many ways have been proposed to estimate task weights before or during training they typically rely on heuristics or extensive search of the weighting space. We propose a novel method called $\alpha$-Variable Importance Learning ($\alpha$VIL) that is able to adjust task weights dynamically during model training, by making direct use of task-specific updates of the underlying model's parameters between training epochs. Experiments indicate that $\alpha$VIL is able to outperform other Multitask Learning approaches in a variety of settings. To our knowledge, this is the first attempt at making direct use of model updates for task weight estimation.

## 1 Introduction

In Machine Learning, we often encounter tasks that are at least similar, if not even almost identical. For example, in Computer Vision, multiple datasets might require object segmentation or recognition (Deng et al., 2009; LeCun et al., 1998; Lin et al., 2014) whereas in Natural Language Processing, tasks can deal with sentence entailment (De Marneffe et al., 2019) or paraphrase recognition (Quirk et al., 2004), both of which share similarities and fall under the category of Natural Language Understanding.

Given that many such datasets are accessible to researchers, a naturally emerging question is whether we can leverage their commonalities in training setups. Multitask Learning (Caruana, 1993) is a Machine Learning paradigm that aims to address the above by training a group of sufficiently similar tasks together. Instead of optimizing each individual task's objective, a shared underlying model is fit so as to maximize a global performance measure, for example a LeNet-like architecture (LeCun et al., 1998) for Computer Vision, or a Transformer-based encoder (Vaswani et al., 2017) for Natural Language Processing problems. For a broader perspective of Multitask Learning approaches, we refer the reader to the overviews of Ruder (2017); Vandenhende et al. (2020).

In this paper we introduce $\alpha$VIL, an approach to Multitask Learning that estimates individual task weights through direct, gradient-based metaoptimization on a weighted accumulation of task-specific model updates. To our knowledge, this is the first attempt to leverage task-specific model deltas, that is, realized differences of model parameters before and after a task's training steps, to directly optimize task weights for target task-oriented multitask learning. We perform initial experiments on multitask setups in two domains, Computer Vision and Natural Language Understanding, and show that our method is able to successfully learn a good weighting of classification tasks.

## 2 Related Work

Multitask Learning (MTL) can be divided into techniques which aim to improve a joint performance metric for a group of tasks (Caruana, 1993), and methods which use auxiliary tasks to boost the performance of a single *target* task (Caruana, 1998; Bingel & Søgaard, 2017).

Some combinations of tasks suffer when their model parameters are shared, a phenomenon that has been termed *negative transfer*. There have been efforts to identify the cause of negative transfer. Du et al. (2018) use negative cosine similarity between gradients as a heuristic for determining negative transfer between target and auxiliary tasks. Yu et al. (2020) suggest that these *conflicting* gradients are detrimental to training when the joint optimization landscape has high positive curvature and there is a large difference in gradient magnitudes between tasks. They address this by projecting task gradients onto the normal plane if they conflict with each other. Wu et al. (2020) hypothesize that the degree of transfer between tasks is influenced by the alignment of their data samples, and propose an algorithm which adaptively aligns embedded inputs. Sener & Koltun (2018) avoid the issue of negative transfer due to competing objectives altogether, by casting MTL as a Multiobjective Optimization problem and searching for a Pareto optimal solution.

In this work, we focus on the target task approach to Multitask Learning, tackling the problem of auxiliary task selection and weighting to avoid negative transfer and maximally utilize positively related tasks. Auxiliary tasks have been used to improve target task performance in Computer Vision, Reinforcement Learning (Jaderberg et al., 2016), and Natural Language Processing (Collobert et al., 2011). They are commonly selected based on knowledge about which tasks *should* be beneficial to each other through the insight that they utilize similar features to the target task (Caruana, 1998), or are grouped empirically (Søgaard & Goldberg, 2016). While this may often result in successful task selection, such approaches have some obvious drawbacks. Manual feature-based selection requires the researcher to have deep knowledge of the available data, an undertaking that becomes ever more difficult with the introduction of more datasets. Furthermore, this approach is prone to failure when it comes to Deep Learning, where model behaviour does not necessarily follow human intuition. Empirical task selection, e.g., through trialling various task combinations, quickly becomes computationally infeasible when the number of tasks becomes large.

Therefore, in both approaches to Multitask Learning (optimizing either a *target task using auxiliary data* or a *global performance metric*), automatic task weighting during training can be beneficial for optimally exploiting relationships between tasks.

To this end, Guo et al. (2019) use a two-staged approach; first, a subset of auxiliary tasks which are most likely to improve the main task's validation performance is selected, by utilizing a Multi-Armed Bandit, the estimates of which are continuously updated during training. The second step makes use of a Gaussian Process to infer a mixing ratio for data points belonging to the selected tasks, which subsequently are used to train the model.

A different approach by Wang et al. (2020) aims to directly differentiate at each training step the model's validation loss with respect to the probability of selecting instances of the training data (parametrised by a scorer network). This approach is used in multilingual translation by training the scorer to output probabilities for all of the tasks' training data. However, this method relies on noisy, per step estimates of the gradients of the scorer's parameters as well as the analytical derivation of it depending on the optimizer used. Our method in comparison is agnostic to the optimization procedure used.

Most similarly to our method, Sivasankaran et al. (2017) recently introduced Discriminative Importance Weighting for acoustic model training. In their work, the authors train a model on the CHiME-3 dataset (Barker et al., 2015), adding 6 artificially perturbed datasets as auxiliary tasks. Their method relies on estimating model performances on the targeted validation data when training tasks in isolation, and subsequently using those estimates as a proxy to adjust individual task weights. Our method differs from this approach by directly optimizing the target validation loss with respect to the weights applied to the model updates originating from training each task.

## 3 $\alpha$-VARIABLE IMPORTANCE LEARNING

The target task-oriented approach to Multitask Learning can be defined as follows. A set of classification tasks $T = \{t_1, t_2, \ldots, t_n\}$ are given, each associated with training and validation datasets, $\mathcal{D}_{t_i}^{train}$ and $D_{t_i}^{dev}$, as well as a *target task* $t^* \in T$. We want to find weights $\mathrm{W} = \{\omega_1, \omega_2, \ldots, \omega_n\}$ capturing the *importance* of each task such that training the parameters $\theta$ of a Deep Neural Network on the weighted sum of losses for each task maximizes the model's performance on the target task's development set:

$$\theta^* = \arg\min_{\theta} \sum_{i=0}^{|T|} \frac{w_i}{\sum w} \cdot \mathcal{L}^{t_i}(\mathcal{D}_{t_i}^{train}, \theta) \quad s.t. \quad \mathcal{L}^{t^*}(\mathcal{D}_{t^*}^{dev}, \theta^*) \approx \min_{\theta} \mathcal{L}^{t^*}(\mathcal{D}_{t^*}^{dev}, \theta) \qquad (1)$$

where $\mathcal{L}^{t_i}(\mathcal{D}_{t_i}, \theta)$ is defined as the average loss over all data points in the dataset $\mathcal{D}_{t_i}$ for network parameters $\theta$, computed using the appropriate loss function for task $t_i$ (in our case the standard cross entropy loss):

$$\mathcal{L}^{t_i}(\mathcal{D}_{t_i}, \theta) = \frac{1}{|\mathcal{D}_{t_i}|} \sum_k \mathcal{L}(x_k, y_k; \theta), \quad (x_k, y_k) \in \mathcal{D}_{t_i} \qquad (2)$$

The introduction of task weights $w$ in Equation 1 aims at scaling the tasks' model updates in a way that positive transfer towards the target is exploited and negative transfer avoided. It is crucial therefore to have an efficient and reliable way of estimating the influence of tasks on the target's performance, and of adjusting the weights accordingly during model training.

To this end, we introduce $\alpha$-Variable Importance Learning ($\alpha$VIL), a novel method for target task-oriented Multitask training, outlined in Algorithm 1. $\alpha$VIL introduces a number of additional parameters – $\alpha$-variables – into the model, which are associated with the *actually realized* task-specific model updates.

---

**Algorithm 1:** The $\alpha$-Variable Importance Learning algorithm.

**Data**: Model parameters $\theta$; a set of tasks $T = \{t_1, \ldots, t_n\}$; a target task $t^*$; training data $D_{t_i}^{train}$ for each task $t_i$; development data $D_{t^*}^{dev}$ for the target task; maximum number of training epochs $\mathcal{E}$; ratio $\rho$ of tasks' training data to sample per epoch; number of $\alpha$ tuning steps $s$

**Result**: updated parameters $\theta$, optimized for performance on $t^*$

1. W $\leftarrow \{w_{t_i} = 1 \,|\, t_i \in T\}$   // initialize all task weights to 1
2. **for** $\epsilon = 1 \ldots \mathcal{E}$ **do**
3.    **for** $t_i \in T$ **do**
4.       $\mathcal{D}_{t_i}^{\epsilon} \overset{\rho}{\sim} \mathcal{D}_{t_i}^{train}$   // sample task-specific data
5.       $\theta_{t_i} \leftarrow \arg\min_{\theta'} \frac{w_{t_i}}{\sum w} \mathcal{L}^{t_i}(\mathcal{D}_{t_i}^{\epsilon}, \theta')$   // task's model update starting at $\theta$
6.       $\delta_{t_i} \leftarrow \theta_{t_i} - \theta$
7.
      // task-specific weight update, optimizing wrt. $\alpha$ parameters on $\delta$
8.    **for** $s \in 1 \ldots s$ **do**
9.       $\{\alpha_1, \alpha_2, \ldots, \alpha_{|T|}\} \leftarrow \arg\min_{\{\alpha_1, \alpha_2, \ldots, \alpha_{|T|}\}} \mathcal{L}(\mathcal{D}_{t^*}^{dev}, \ \theta + \alpha_1\delta_1 + \ldots + \alpha_{|T|}\delta_{|T|})$
10.    $\theta \leftarrow \theta + \alpha_1\delta_1 + \ldots + \alpha_{|T|}\delta_{|T|}$
11.    W $\leftarrow \{w_{t_i} + (\alpha_{t_i} - 1) \,|\, t_i \in T\}$

---

During training, our approach first performs weighted task-specific model updates on a proportion of the available training data for each individual task, starting from the current model parameters. It collects the resulting model deltas, i.e., the differences between the model's parameters before and after the singletask update and resets the model. After this *delta collection phase*, the optimal mixing factors, that is, $\{\alpha_1, \alpha_2, \ldots, \alpha_{|T|}\}$ of the model updates are found, such that the parameters resulting from the interpolation of scaled task-specific $\delta$'s minimize the loss on the target task's development data.

The $\alpha$–parameters can be optimized through any type of optimization method however, since our models are end to end differentiable, we can backpropagate directly and use gradient descent.

Once we have found the optimal mixing ratio of task updates, we write the new state back to the model, and update the task weights subject to the optimized $\alpha$ parameters.

The task weight update rule (line 11 in Algorithm 1) combined with the weighted task-specific model updates (line 5) tries to capture the intuition that if a task update was up- or down-scaled in the $\alpha$-tuning stage, we likely want to update the parameters more/less for this task, during the next delta collection phase.

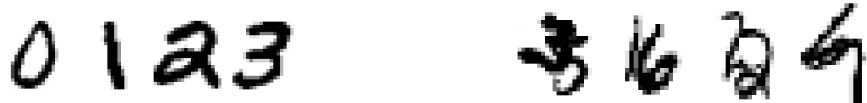

Figure 1: Examples images of digits found in MNIST (left) and the super-imposed digits present in MultiMNIST (right). Gold labels for the MultiMNIST examples for top-left/bottom-right respectively are 3/5, 1/6, 2/2, 6/7.

## 4 EXPERIMENTS

To test the efficacy of $\alpha$VIL, we apply it in two domains, Computer Vision (CV) and Natural Language Processing (NLP). We perform experiments on a multitask version of the MNIST dataset (LeCun et al., 1998), and on a number of well-established Natural Language Understanding (NLU) tasks. In all scenarios, we evaluate $\alpha$VIL against baselines that perform single task and standard multitask learning, as well as against a strong target task-oriented approach.

### 4.1 COMPUTER VISION

As the first benchmarking domain for $\alpha$VIL we chose Computer Vision, as Multitask Learning has a longstanding tradition in this field, with a variety of existing datasets. For our experiments, we use Sener & Koltun (2018)'s variation of MultiMNIST, itself introduced in Sabour et al. (2017) as an augmentation of the well-established MNIST dataset. In MNIST, the task is to classify a given hand-drawn digit (Figure 1, left). For MultiMNIST, two digits are overlaid, the first shifted to the top left and the second to the bottom right (Figure 1, right). The two tasks are to recognize each of the super-imposed digits. The resulting MultiMNIST dataset contains a total of 10.000 test and 60,000 training instances, of which we sample 10,000 at random to use for validation.

For all Multitask experiments on MultiMNIST, we use a classification architecture similar to Sener & Koltun (2018), as depicted in Figure 2. The model comprises a shared convolutional encoder, where the first convoluational layer has 10 filters with kernel size 5 and the second uses the same kernel size but 20 filters. Both max pooling layers are of size 2x2, and the shared fully connected layer is of dimensionality 320x50. The encoded images are passed into task-specific heads of size 50x10, used to classify the top-left and bottom-right digit respectively.

We compare $\alpha$VIL to a single task baseline, where we remove one of the classification heads from the model, as well as a standard multitask baseline in which both heads are trained jointly, each of which receives the shared encoder output for a given image and updates the model to solve its specific task, averaging their updates for the image. For the standard multitask baseline, we save two snapshots of the model during training, each of which performs best on either the top-left or the bottom-right digit on the development sets. For $\alpha$VIL, we predefine either one of these tasks as the target and try to optimize its performance using the other task as auxiliary data. We also compare our method to the Discriminative Importance Weighting (DIW) approach of Sivasankaran et al. (2017), which provides a very strong target task-oriented optimization method. Their approach is relatively similar to our own however, in contrast to $\alpha$VIL, DIW collects unweighted single-task updates, and evaluates each individual task on the specified target task. It then performs actual weighted multitask updates in an inner loop, evaluates the jointly updated model, and adjusts individual task weights with respect to the difference in evaluation results, after which the model is reset and re-trained with

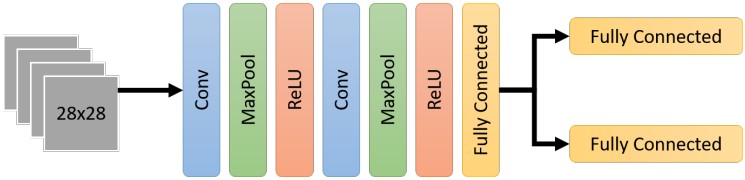

Figure 2: General Multitask model architecture for MultiMNIST experiments.

| | Development Set Accuracy | | | | | | | |
| | Task 1 (Top-Left Digit) | | | | Task 2 (Bottom-Right Digit) | | | |
| | Min | Max | Mean | Std. | Min | Max | Mean | Std. |
|---|---|---|---|---|---|---|---|---|
| Singletask | 96.12 | **97.25** | 96.73 | 0.30 | **94.95** | **96.48** | 95.57 | 0.46 |
| Multitask | 95.99 | 96.71 | 96.37 | **0.20** | 94.43 | 95.54 | 95.00 | 0.33 |
| DIW | 95.94 | 97.14 | 96.84 | 0.27 | 94.83 | 96.32 | 95.60 | 0.45 |
| $\alpha$VIL | **96.18** | 97.24 | **96.91** | 0.25 | 94.78 | 96.22 | **95.89** | **0.30** |
| | Test Set Accuracy | | | | | | | |
| Singletask | **95.90** | 96.89 | 96.44 | 0.35 | 93.97 | 95.48 | 94.83 | 0.44 |
| Multitask | 95.40 | 96.29 | 95.92 | 0.24 | **94.34** | 95.02 | 94.67 | **0.23** |
| DIW | 95.67 | 96.85 | 96.46 | **0.29** | 94.09 | 95.52 | 94.93 | 0.42 |
| $\alpha$VIL | 95.74 | **96.95** | **96.56** | 0.32 | 94.13 | **95.58** | **95.16** | 0.33 |

Table 1: Model classification accuracy and standard deviation over 20 random seeds, on MultiMNIST development and test sets, for single task and standard multitask baselines, Discriminative Importance Weighting, and $\alpha$VIL. Results in bold indicate best overall approach.

the new weights, until an improvement over the previous iteration's target task performance has been achieved.

For all experiments, we use a batch size of 256 and SGD as the model optimizer with learning rate set to 0.05, momentum to 0.9. Dropout is not used. We train on the entire MultiMNIST training set before evaluating (single task, multitask) or weight tuning on the development data (DIW, $\alpha$VIL). For $\alpha$VIL, we set the number of tuning steps $s = 10$, and use SGD with learning rate set to 0.005 and momentum to 0.5 as the meta-optimizer to tune the $\alpha$ parameters. The delta collection weights, $w_{t_i}$, as well as the task importance weights of DIW are clamped to $[10^{-6}, \infty)$[1]. Similarly, we set an early stopping criterion for the re-weighting loop of DIW to break after 10 epochs without improving over the previous performance[2]. We train all models for 100 episodes, keeping the best-performing snapshots with respect to the development accuracy and target task[3]. We average over 20 random seeds, and report in Table 1 the minimum, maximum and mean classification accuracy on the MultiMNIST development and test sets, as well as the models' standard deviation.

We can see from Table 1 that the second task, classifying the bottom-right digit, seems to be somewhat harder for the model to learn than the first, as reflected by the worse model performance in all settings for this task. Furthermore, when training in the standard multitask setting, model performance actually *decreases* for both tasks. This indicates that in MultiMNIST, there exists *negative transfer* between the two tasks. We can also see that both target task-oriented Multitask approaches are able to rectify the negative transfer problem, and in fact even improve over the models trained on the single tasks in isolation.

Across the board, $\alpha$VIL achieves the best overall mean accuracy, on both tasks' development and test sets, while keeping a comparably low standard deviation. Crucially, $\alpha$VIL not only brings Multitask performance back to the single task level, but outperforms the single task baseline, as well as the DIW target task-oriented training approach.

Figure 3 shows the $\alpha$–parameters over the course of training (left) and the corresponding normalized weights of the two tasks (right). The target task $t^*$ is set to task 1. As expected, due to the existence of negative transfer between the tasks, the algorithm initially weights model updates originating from the main task and auxiliary task with $\alpha_1 > 1$ and $\alpha_2 < 1$ respectively, quickly driving the task weights down. Finally, $\alpha$'s settle at $\approx 1$, as task 2 has been essentially excluded from the training.

---

[1] In practice, training with even a slightly negative loss weight causes parameters to massively overshoot.

[2] In their original paper, the authors mention no such early stopping criterion however, we found that training can enter an infinite loop without it.

[3] For standard multitask training, we save 5 snapshots of the model, one for each best performance on the individual tasks

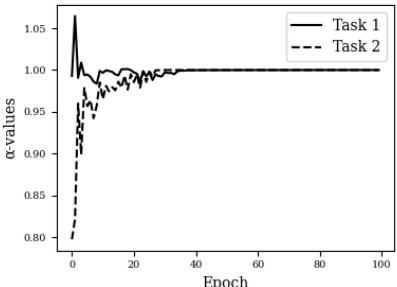 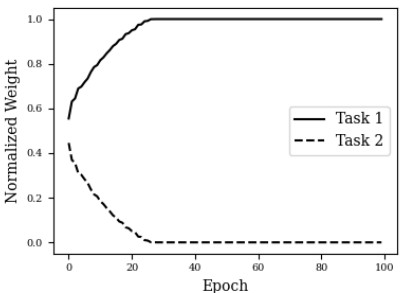

Figure 3: $\alpha$ parameter values (left) and task-specific weights (right) over the course of $\alpha$VIL training on MultiMNIST $\alpha$VIL. Task 1 is the target task.

## 4.2 Natural Language Understanding

We also test $\alpha$VIL in the domain Natural Language Processing. In particular, we are interested in the learning of Natural Language Understanding (NLU) tasks.

The field of NLU has recently regained traction as a research area, with the introduction of the Transformer architecture (Vaswani et al., 2017) and subsequent advent of very large language models such as BERT (Devlin et al., 2019) and similar models. These very deep models are trained on a plethora of data, and (at least seem to) incorporate a large amount of linguistic knowledge (Tenney et al., 2019; Staliūnaitė & Iacobacci, 2020), which makes them ideal for downstream tasks like NLU.

Natural Language Understanding comprises a wide variety of established datasets and tasks, each dealing with different aspects of the broader field. This provides a rich and interesting resource for Multitask Learning research, as NLU tasks are at least at first glance related in the kind of linguistic knowledge they require. They therefore lend themselves to being trained jointly, yet tasks might in reality make use of different aspects of the underlying shared model, leading potentially to negative transfer.

For a first test of $\alpha$VIL in the NLU domain, we limit our experiments to 5 commonly used tasks that are also represented in the GLUE and SuperGLUE (Wang et al., 2019a;b) research benchmarks: CommitmentBank (De Marneffe et al., 2019, CB), Choice of Plausible Alternatives (Roemmele et al., 2011, CoPA), Microsoft Research Paraphrase Corpus (Quirk et al., 2004, MRPC), Recognizing Textual Entailment (Dagan et al., 2006; Bar Haim et al., 2006; Giampiccolo et al., 2007; Bentivogli et al., 2009, RTE), and Winograd Natural Language Inference (WNLI), itself based on the Winograd Schema Challenge (Levesque et al., 2011). Brief descriptions and examples for each of the tasks are given below. The tasks were chosen for their relatively small data sizes, to allow for faster experimentation.

**CommitmentBank:** CB is a three-way classification task, where model inputs consist of a *premise* text of one or more sentences, e.g, *"B: Oh, well that's good. A: but she really doesn't. Nobody thought she would adjust"*, and a *hypothesis* like *"she would adjust"* which can either be entailed in the premise, contradict it, or be neutral. CB is the smallest of our NLU tasks, and contains 250 examples of training data, as well as 56 for validation and 250 for testing.

**CoPA:** This task provides a *premise* (*"My body cast a shadow over the grass."*), two alternative *choices* (*"The sun was rising."*/*"The grass was cut."*), and a *relation* (*"cause"*) as input. The model's task is to determine whether *choice1* or *choice2* is the more likely continuation of the premise, given the relation. The CoPA dataset provides 400 training as well as 100 and 500 validation and test instances.

**MRPC:** This is a paraphrase recognition task, in which the model is shown two *sentences*, and to classify whether or not they are *paraphrases*. MRPC constitutes our largest dataset for NLU experiments, with 3668 training and 408 validation examples, with 1725 for testing.

**Recognizing Textual Entailment:** RTE is a two-way classification task, where given a *premise*, for example *"Things are easy when you're big in Japan."* and a *hypothesis*, such as *"You're big*

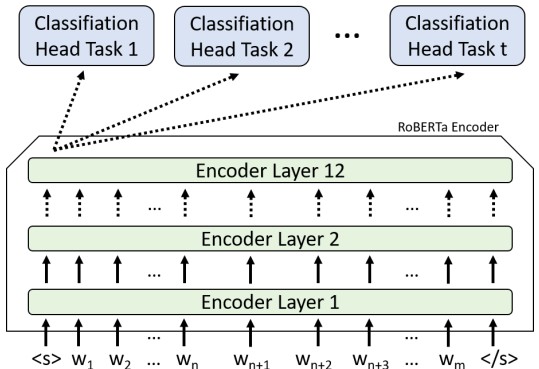

Figure 4: General Multitask model architecture for NLU experiments.

*in Japan."*, the task is to correctly classify whether the former entails the latter. The RTE data is our second larges dataset, and contains 2490 instances for training, as well as 277 and 3000 for validation and testing respectively.

**WNLI:** This task is about classifying Natural Language Inference. The input consists of a short *text* (*"I stuck a pin through a carrot. When I pulled the pin out, it had a hole."*) and a follow-up *sentence* (*"The carrot had a hole."*), and the model has to correctly classify whether the sentence can be inferred from the preceding text. WNLI comprises training, validation, and test sets containing 635, 71, and 146 instances, respectively.

To address the Multitask learning problem on these 5 NLU tasks, similar to the MultiMNIST image classification setup above, we employ an architecture of joint encoder and task-specific heads which is depicted in Figure 4. We use a RoBERTa model (Liu et al., 2019) as the underlying shared encoder, and prepare the data for all tasks such that is suitable as Transformer encoder input. For our experiments, we employ the pre-trained RoBERTa$_{base}$ encoder provided by Huggingface[4], which consists of 12 encoder layers, and outputs a 768-dimensional embedding vector for each input token. On top of the shared encoder we add one classification head for each of the 5 NLU tasks. Each head takes as input the final RoBERTa encoding of the *start-of-sequence* token $$, and employs a linear layer of dimensionality 768x256, ReLU, followed by another linear layer into the classification space (2 output units for CoPA, MRPC, RTE, and WNLI, 3 output units for CB).

As for the Computer Vision experiments above, we perform experiments on the 5 NLU tasks with standard single task and multitask baselines, Discriminative Importance Weighting, and $\alpha$VIL. All experiments use the same model architecture, except in the single task setup where all heads but that for the target task are disabled. We use AdamW (Loshchilov & Hutter, 2017) as the optimizer for the base model, with a learning rate of $5e^{-6}$, $\epsilon = 1e^{-6}$, and weight decay of $0.01$. For DIW, we use a weight update learning rate of $0.1$, and we use SGD with a learning rate of $0.001$ and momentum of $0.5$ for $\alpha$-variable optimization. Parallel to the previous experiments, we use 10 $\alpha$ update steps for $\alpha$VIL, and an early stopping patience of 10 for DIW. Other than for MultiMNIST, where both tasks share a common input, each NLU task comes with distinct model inputs and outputs, as well as with datasets of very different sizes. We therefore do not wait for an entire episode before model evaluation, but instead increase the evaluation frequency. This is necessary as some of the tasks tend to overshoot their optimal point during training, and we might miss the true best model performance if we wait for an entire episode to finish. To increase the evaluation frequency, we sample at each epoch $25\%$ of the respective training sets, train the model on batches of this quarter of the total data, and evaluate. For DIW and $\alpha$VIL we also use this $25\%$ split for weight adjustment. The batch size is set to 8 for all tasks, and we train for a total of 20 epochs, keeping the best performing snapshot per task.

---

[4] https://huggingface.co/roberta-base

|  | CB | | CoPA | | MRPC | | RTE | | WNLI | |
|---|---|---|---|---|---|---|---|---|---|---|
| Singletask | 90.18 | 84.0 | 56.00 | 47.6 | 90.01 | 87.8 | 80.23 | 73.1 | 55.99 | **65.1** |
| Multitask | 95.98 | **94.0** | **68.00** | **60.0** | 89.83 | 87.8 | 80.05 | 75.3 | 56.69 | **65.1** |
| DIW | **98.66** | 93.6 | 64.25 | 56.4 | **90.69** | **88.2** | **81.14** | 75.1 | **59.15** | 62.3 |
| $\alpha$VIL | 97.77 | 92.8 | 67.33 | 59.0 | 89.22 | **88.2** | 81.05 | **75.8** | 57.75 | **65.1** |

Table 2: Average classification accuracy on the 5 tested NLU development datasets over 4 random seeds per task (left column per task), as well as GLUE / SuperGLUE test accuracy for ensembles (right column), for single task and standard multitask baselines, Discriminative Importance Weighting, and $\alpha$VIL. Results in bold indicate best overall training approach. For space constraints, we show only average development accuracy and final enemble test scores.

Table 2 summarizes the results of the NLU experiment, with model accuracies on the tasks' development sets over 4 random seeds. To obtain test set accuracies, we submit for each method the predictions of ensembles of the trained models to the GLUE and SuperGLUE benchmarks[5].

Even though these experiments were carried out on a very limited number of tasks and random seeds, and should accordingly be interpreted with care, it is clear that Discriminative Importance Weighting constitutes a very strong Multitask Learning system for comparison, especially with respect to development set performance.

This should be of little surprise: First, DIW does optimize its task weights *directly* based on the target task's development accuracy, re-weighting and re-training at each epoch until a new highest score is found. This is in contrast to $\alpha$VIL, which performs a predetermined number of optimization steps on the average development set loss, and just one interpolation of model updates per epoch. Second, as there is no perfect correlation between a lower *average* loss and higher absolute accuracy – decreasing the first slightly is not guaranteed to immediately increase the second – $\alpha$VIL might not find a new best development accuracy each epoch.

However, the picture changes when considering actual performance on unseen test data. While DIW almost consistently performs best on the development data, $\alpha$VIL actually outperforms it on the final test scores. For 3 out of the 5 tested NLU tasks, the $\alpha$VIL ensembles are ranked first on test (shared with DIW on MRPC, and singletask and standard multitask on WNLI). For one more dataset (CoPA), $\alpha$VIL comes second, trailing the best system by just 1 point.

We conjecture that DIW is more prone to overfitting than $\alpha$VIL, precisely because of DIW's heavy reliance on tuning on dev multiple times each epoch until finding a new best performance. This might be less severe in settings where training, development, and test datasets are both large and sufficiently similar, as is the case in the MultiMNIST experiment above where no such performance discrepancies seem to manifest. However, in the NLU domain with less data and potentially large difference between training, development, and test examples, overfitting constitutes a more severe problem, which $\alpha$VIL seems to be able to better avoid.

## 5 CONCLUSIONS AND FUTURE WORK

In this work we have introduced $\alpha$VIL, a novel algorithm for target task-oriented multitask training. $\alpha$VIL uses task-specific weights which are tuned via metaoptimization of additional model parameters, with respect to target task loss. Experiments in two different domains, Computer Vision and NLP, indicate that $\alpha$VIL is able to successfully learn good task weights, and can lead to increased target-task performance over singletask and standard multitask baselines, as well as a strong target task-oriented optimization approach.

$\alpha$VIL's formulation is very flexible and allows for many variations in its (meta)optimization approach. In the future, we would like to experiment more with different ways to optimize $\alpha$ parameters than standard SGD. Also, $\alpha$VIL does not currently perform a joint optimization iteration after its $\alpha$ estimation task and re-weighting, which could lead to further performance gains.

---

[5] https://gluebenchmark.com; https://super.gluebenchmark.com/

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
