# OpenReview forum: "$\alpha$VIL: Learning to Leverage Auxiliary Tasks for Multitask Learning"
_ICLR.cc/2021/Conference — Reject_

### Official Review · AnonReviewer1 · 2020-10-27
**Multi-task learning with gradient-based meta-optimization for learning task-specific weights**

**Rating:** 3
**Confidence:** 4

**Review:**

Summary:
This paper proposes a new approach for multi-task learning that estimates the individual task weights through gradient-based meta-optimization on a weighted accumulation of task-specific model updates. Evaluations are performed in a multi-task learning setup on tasks related to computer vision (Multi-MNIST) and natural language understanding (tasks from GLUE and SuperGLUE).

Pros:
1) The paper is easy to follow.

2) Empirical evaluation is performed on vision and NLU domains.

Cons:
1) I am not completely convinced with the proposed alpha-Variable Importance Learning algorithm. It is not very clear in the discussion how the alpha is different from task-specific weights. For example, in algorithm-1, if you replace deltas in line-10 with line-6, then there is no need to have separate alphas and task-specific weights, where line-9 can calculate the task-specific weights directly.

2) In general, for a multi-task setup, I would expect to show the multi-task learning with multiple auxiliary tasks (that’s the main motivation of this paper as well). However, the choice of the experimental setup is convincing, especially for the vision domain there is only one auxiliary task.

3) Both results in Table-1 and Table-2 suggest that the proposed algorithm is not superior over the baselines and previous approaches. The improvements are minor and sometimes lower, and I believe most of the results fall within the statistically insignificant range.


Overall:
I think the paper can be made stronger with more thorough discussion on the algorithm and its properties. Further, the experimental results suggest that the proposed algorithm performs more or less similar to previous methods. Hence, there is a lot of scope for further improvement and I would suggest rejecting this paper. I would also suggest the authors to perform more experiments and ablations.


Questions:
1) How is the alpha different from task-specific weights. Please discuss more on this. In algorithm-1, if you replace deltas in line-10 with line-6, then there is no need to have separate alphas and task-specific weights?


2) Please provide statistical significant scores for all the results.

3)  What's the reason behind choosing a multi-MNIST dataset with only one auxiliary task? Aren’t there other datasets in a MTL setup with more auxiliary tasks?

4) Table-2 results for the development set are based on the average of multiple runs, but for test you reported the ensemble, so why don’t you report ensemble for the development set as well?

5) Can you also present some ablations/discussion on the learned importance of an auxiliary task (based on task-specific weight over the training trajectory) vs. any intuitive reason that makes sense of this importance of the axillary task for a given primary task? If there is not such correlation, then also it's good to discuss.

Other comments:

1) Please try to expand the introduction section.

2) Please provide some more ablations.

---

> ### Author Response · Authors · 2020-11-24
> **Review-specific Response**
>
> We would like to thank you, your comments are very helpful in our efforts to improve our work.
> Please refer to our general comment, where we believe we answered the points raised in "Cons" and "Overall".
>
> For specific questions:
> The task weights are there to 'accumulate' in time the relative task weighing that has been calculated using the alpha optimization. The intuition is that if a task's change in the model has been always applied with a weight < 1, we probably want to collect future changes by this task with a scale in its gradients that reflects this downweighting. However, it is correct that it's possible for a single metaparameter to decide at the end how to weight a task delta without the need for extra scaling during collection. We have not done any experiments to compare the two methods yet but it'd be a good addition for an updated version of the paper.
>
> We chose MultiMNIST for two reasons. First, we knew that multitask learning for this dataset helps, as this had been established in prior work (in particular, the MultiMNIST paper itself). Second, as the tasks overlap in terms of their classification space, but concentrate on different parts of the image, we could reasonably expect the task weights to eventually go towards 1.0 and 0.0 for the main and auxiliary tasks respectively (see also Figure 3, which shows they actually do). This would add as a sort of sanity check. We should definitely have made this point clearer in the paper, and will do so in the updated version.
>
> Ensembles were chosen for the test set as GLUE and superGLUE only allow a very limited number of submissions to be tested, leaving us the choice between ensemble or single-best models per method. While we believe that average scores per methods are more meaningful, we agree that we should also include ensemble results for the development set to be consistent with the test setting.

---

### Official Review · AnonReviewer3 · 2020-10-29
**Ad hoc multitask learning via task weighting with unconvincing experiments**

**Rating:** 4
**Confidence:** 4

**Review:**

Summary: This paper presents an algorithm for multitask learning that learns task weights via an EM-like approach that alternates between updating the model parameters (using task weights) and updating the task weights (using current model parameters, based on the target task development set).

Experiments: They compare against single task training, standard multitask training (though this isn’t described very clearly but roughly is training jointly on tasks), and another method for learning task weights, Discriminative Importance Weighting (DIW).
They present experiments on MultiMNIST, where the two tasks are to predict the task in the top left and bottom right of two superimposed digits. The proposed algorithm has the beast mean performance, but results are within a standard deviation of the baselines. They also present experiments on 5 NLU tasks (CommitmentBank, COPA, MRPC, RTE, WNLI) with the same baselines. The results on these tasks are mixed, with all multitask methods outperforming single task training (except on WNLI, which is a bit degenerate).

Overall, this paper needs a bit more work. The proposed is quite ad hoc, and with little justification, it’s not clear why we should be doing any of the things the algorithm proposes. For example, in line 11 of Algorithm 1, I don’t understand the intuition arbitrarily subtracting 1 from all weights. From a novelty perspective, I’m not convinced the proposed method is different enough from existing methods. Dynamic task weighting is not particularly new (e.g. the baseline method, DIW, they compare against), and their method starts to look a lot like meta-learning of task weights (like MAML [1], [2], or [3]). The results from the experiments are not convincing to me. On MNIST, the results between all methods are fairly close together, and on the NLU tasks, there’s no clear best algorithm.

Additional notes and questions
1. For the “standard multitask baseline”, are the tasks balanced in size? Do you deterministically train on a batch from both or is it stochastic? This is mostly relevant for the NLU tasks, which have fairly different sizes.
2. On MNIST, given that the algorithm sets the weight of one task to 1.0 and the other to 0.0, why is this algorithm outperforming single-task training?
3. On a similar note, it'd be nice to see a sanity check experiment that the learned weights are sensible (e.g. one task has random labels) or an examples of where the learned weights are binary.
4. I appreciate that the authors report min/max/mean/std of 20 runs on MNIST. It would be nice to see the standard deviations for the NLU tasks for consistency and given the fact that the standard deviations on the MNIST task were important in differentiating significant differences. Similarly, it would be nice to see how the task weights evolve.

Style notes
* Huggingface Transformers now has a citation
* Multitask Learning; Computer Vision; Natural Language Processing/Understanding: lowercase
* “Singletask” should probably be hyphenated or two words.
* “10.000” → “10,000”
* Table 2 could really use headers over the two columns within each task.

[1] Finn, Chelsea, Pieter Abbeel, and Sergey Levine. "Model-Agnostic Meta-Learning for Fast Adaptation of Deep Networks." ICML. 2017.
[2] Shu, Jun, et al. "Meta-weight-net: Learning an explicit mapping for sample weighting." Advances in Neural Information Processing Systems. 2019.
[3] Wang, Xinyi, Yulia Tsvetkov, and Graham Neubig. "Balancing training for multilingual neural machine translation." arXiv preprint 2004.06748 (2020).

---

> ### Author Response · Authors · 2020-11-24
> **Review-specific Response**
>
> Thank you for your helpful comments.
>
> In addition to our general response, we would like to answer your specific questions here.
>
> "For example, in line 11 of Algorithm 1, I don’t understand the intuition arbitrarily subtracting 1 from all weights." -- We see how this is unclear and should be pointed out in the text. We are not subtracting 1 from all weights, but from the newly found alpha parameters. Alpha parameters tell us which way to adjust the weights, i.e., whether to increase or decrease a task's importance. Since alphas are initialised to 1 before being optimised to weigh the tasks (lines 8--10, Algorithm 1), an alpha-value >1 entails that the corresponding task importance weight should be greater, while an optimized alpha < 1 indicates that the task should be down-weighted. Accordingly in line 11 of Algorithm 1, the term (\alpha - 1) will be positive if alpha > 1 and the new weight according to w+(\alpha-1) will be increased. Conversely, if alpha < 1, the overall term will be negative, and thus substracted from the task weight, decreasing it.
>
> On the novelty of our method, while dynamic task weighting is not new, we believe that determining task weights through direct metaoptimization is. While DIW aims to optimize the weights with a numerical estimate of the gradient, we see our work as a more general framework which is compatible with any optimization method (e.g. Adam). We see how this could seem similar to meta-learning algorithms like MAML or Reptile. However, the metalearning objective is different i.e., instead of searching for a model initialization that can be used for rapid finetuning, we are looking for task weights during training, skipping this step. Other Meta-weight approaches like Shu et al. (2019) do adjust sample-specific weights for a single task during training by learning a complex weighting function. This differs from our approach as we are looking at data accross different tasks rather than within the same task, and our weighting is a very simple interpolation step of different task-specific updates.
>
> Additional notes/questions:
>
> 1. In NLU, for standard multitask, due to the fact that task datasets are not balanced, we sample 25% of all data for each task and train with this to calculate a task delta.
>
> 2. While the weights eventually go to 1.0 and 0.0 respectively, we observed in Figure 3 that they do so gradually over the course of training to about epoch 25. We conjecture that initially, there is at least some benefit conveyed by the auxiliary task, which has implications for the final trained model.
>
> 3. In part, MultiMNIST was meant to provide this sanity check, in combination with Figure 3. We should have made this more clear in the text. We tried more sanity checks, e.g., splitting the datasets into parts and looking to see if the algorithm will pick the same task's splits (that are guaranteed to positively transfer), but we were short of space in the paper.
>
> 4. We can add standard deviations for NLU in the camera ready version however, for the submitted version they are collected over only 4 random seeds, so are less meaningful than for MNIST where they are based on a much larger 20 runs.

---

### Official Review · AnonReviewer4 · 2020-10-29
**The proposed methodology is intuitive and flexible, the experimental results are not convincing enough.**

**Rating:** 4
**Confidence:** 4

**Review:**

This paper proposes a novel multi-task learning method which adjusts task weights dynamically during training, by exploiting task-specific updates of the model parameters between training epochs. Specifically, the proposed model takes the differences between the model’s parameters before and after the singletask update, after that the mixing factors of the model updates are found based on the differences to minimize the loss on the target task’s development data. Empirical studies are performed on tasks of computer vision and natural language understanding.

The paper is well written and easy to follow, the authors summarize the related work in a clear manner. The proposed methodology is intuitive and well-motivated, in the meantime, it is flexible and can be generalized to other variations in terms of models and tasks.

My major concern about the paper include the following:
1)	Although the proposed method is intuitive and straightforward, it would be necessary to provide some theoretical justification or a formal analysis of the proposed methodology.

2)	Considering the lack of a theoretical justification, the experimental results are not convincing enough to justify the proposed method. The baselines chosen include standard multi-task learning and one single task-oriented approach, which is somewhat limited. Even so, on both of the computer vision and natural language understanding tasks, the proposed method doesn’t consistently outperform the baselines in most cases. The authors did provide sufficient analysis, nevertheless, it doesn’t justify the effectiveness of the algorithm.

Based on the concerns above, the paper can be improved from both the theoretical and empirical perspectives.

---

> ### Author Response · Authors · 2020-11-24
> **Review-specific Response**
>
> Thank you for your feedback and suggestions. We hope we have replied to your concerns in our general response.
>
> We would just like to add that our baselines include standard MT learning, and a single-task oriented approach (each task trained in isolation) as pointed out, as well as Discriminative Importance Weighting. DIW is a very competitive and strong target-task oriented MT algorithm, and in its formulation close to aVIL, with the difference of aVIL tuning task-specific weights through additional optimization. The experiments show aVIL on average outperforms DIW on the tested domains, and in particular, does seem to suffer less from overfitting on the development set(s).

---

### Author Response · Authors · 2020-11-24
**General Response to Reviewers**

We would like to thank the reviewers for their genuinely helpful comments. We are very glad there is consensus that the paper is well written and easy to understand.

We will address issues raised by multiple reviewers here, and reviewer-specific questions in their own comments.

There was a general concern that results are relatively weak and only marginally improve over the compared methods, in particular Discriminative Importance Weighting. We agree that the numbers leave this impression especially in the tested NLP domain.
To put the numbers in perspective, we would like to point out that improvements yielded by multitask learning on NLU tasks are often small. This phenomenon is also observed in other multitask settings, for example in the survey of [1] where MT leads to very mixed results.
Looking at SuperGLUE (of which we used a subset of tasks), the overall average scores of the RoBERTa_large model in single and multitask setup differ by only 1.1 points (due to computational constraints we used the smaller _base variant in this work). Furthermore, on CommitmentBank, CoPA, and RTE the accuracy differences yielded by multitask training are +0.4, +0.6, and -0.1. This goes to show that in general achieving a substantial accuracy improvement is tough with the given model on these tasks.

On the other hand, on the more 'artificial' task of MultiMNIST (less noise, larger, cleaner and more consistent data), aVIL consistently outperforms the compared methods wrt. mean performance, including the very strong DIW which is close in its formulation to aVIL, but relies on a more aggressive weight tuning approach.
On the same note, we believe that one advantage of aVIL over DIW is that is less prone to this overfitting on the development data, as we briefly point in the last paragraph of Section 4.

Another common criticism is a lack of theoretical motivation/justification of the proposed algorithm. We have to concede that the work at present is lacking in this regard, and intend to remedy this for the camera-ready version.

The final common suggestion between reviews was the addition of more experiments and/or ablation studies to show the efficacy of our method. As we were hard-pressed to fit the algorithm along with the existing MultiMNIST and NLP experiments into the page limits, we had to cut out additional experiments and analysis. We will add this to the main text of a camera-ready version, space permitting, or add respective appendices.


[1] Vandenhende et al. (2020) "Multi-Task Learning for Dense Prediction Tasks: A Survey"

---

### Decision · Program_Chairs · 2021-01-07
**Final Decision**

**Decision:**

Reject

**Comment:**

This paper proposes \alphaVIL, a method for weighting the task-specific losses in a multi-task setting in order to optimize the performance on a particular target task. The idea is to first collect gradient updates for the model based on all the separate tasks, and then re-weight those updates in order to optimize the loss on a held-out development set for the target task. In practice, this meta-optimization is performed with gradient descent. Experiments on multi-MNIST and several tasks that are part of GLUE and SuperGLUE show that \alphaVIL is close in performance to a baseline multitask method and discriminative importance weighting.

Strengths:
- The idea is intuitively appealing. Directly reweighting tasks as a meta-optimization step is straightforward and appears to not be proposed previously in the literature.
- The paper is clear in its presentation.

Weaknesses:
- The reviewers agree that the main weakness is that the experimental results do not show that \alphaVIL offers any substantial benefits over existing methods. On the multi-MNIST task, while \alphaVIL tends to have the highest mean performance, the difference is small (less than a standard deviation). On the GLUE/SuperGLUE tasks, it outperforms other methods on only 1 out of 10 experiments. There are also no confidence intervals/standard deviations provided to assess the significance of the results.